# Entropy Generation and Heat Transfer Performance in Microchannel Cooling

**DOI:** 10.3390/e21020191

**Published:** 2019-02-18

**Authors:** Jundika C. Kurnia, Desmond C. Lim, Lianjun Chen, Lishuai Jiang, Agus P. Sasmito

**Affiliations:** 1Department of Mechanical Engineering, Universiti Teknologi PETRONAS, 32610 Bandar Seri Iskandar, Perak Darul Ridzuan, Malaysia; 2State Key Laboratory of Mining Disaster Prevention and Control, Shandong University of Science and Technology, Qingdao 266590, China; 3Department of Mining and Materials Engineering, McGill University, Frank Dawson Adams Bldg., 3450 University Street, Montreal, QC H3A2A7, Canada

**Keywords:** cooling channel, computational fluid dynamics, entropy generation, oblique fin

## Abstract

Owing to its relatively high heat transfer performance and simple configurations, liquid cooling remains the preferred choice for electronic cooling and other applications. In this cooling approach, channel design plays an important role in dictating the cooling performance of the heat sink. Most cooling channel studies evaluate the performance in view of the first thermodynamics aspect. This study is conducted to investigate flow behaviour and heat transfer performance of an incompressible fluid in a cooling channel with oblique fins with regards to first law and second law of thermodynamics. The effect of oblique fin angle and inlet Reynolds number are investigated. In addition, the performance of the cooling channels for different heat fluxes is evaluated. The results indicate that the oblique fin channel with 20° angle yields the highest figure of merit, especially at higher *Re* (250–1000). The entropy generation is found to be lowest for an oblique fin channel with 90° angle, which is about twice than that of a conventional parallel channel. Increasing *Re* decreases the entropy generation, while increasing heat flux increases the entropy generation.

## 1. Introduction

Over the last decades, human dependence on technology, especially electronic devices, has been growing exponentially. In almost every aspect of our daily life, we heavily rely on electronic devices. In order to operate these devices at their optimum performance and achieve their maximum lifespan, maintaining their temperature below the maximum allowable point is crucial. With the advances in semiconductor technology, electronic devices become more powerful but at the same time these devices also generate more heat. Therefore, electronic cooling has been a major concern for research and development with various cooling technologies and approaches have been proposed and evaluated. Some of these cooling approaches are liquid cooling, natural or forced convection cooling, edge cooling, and phase change cooling. Having relatively high cooling capacity and simple arrangement, liquid cooling has been the primary choice to maintain the temperature of high performance electronic devices or components, especially high performance microprocessor chips.

In liquid cooling, two strategies are commonly adopted, i.e., direct cooling where the electronic devices/components is immersed in a cooling liquid (coolant) and indirect cooling where the coolant is driven through a channel attached to the cooled devices/components. The primary advantage of the former is direct contact between coolant and the electronic components may offer better cooling performance. It should be noted, however, that the coolant has to be electrically non-conductive to avoid short circuits, therefore water is not a choice for this application. On the other hand, indirect cooling offer advantages of more flexibility in choices of the coolant and the possibility of using water. The challenge for indirect cooling is to design a cooling channel which offers high cooling performance while maintaining lower pumping power requirement.

Owing to its critical role in determining the cooling performance and pumping power requirement, cooling channel design has been subject to numerous studies. The most widely studied and adopted cooling channels are rectilinear designs: parallel, serpentine and variations thereof. Kurnia et al. [1] numerically evaluated the heat transfer performance of several cooling channel designs: rectilinear, wavy, oblique fin and coiled. The performance was evaluated in terms of figure of merit, i.e., the ratio of heat transfer to the required pumping power. It was found that the parallel channel with oblique fins offers the highest heat transfer per unit pumping power. Wang et al. [2] investigated the effect of cross-section profile in a parallel micro-channel heat sink. Three different cross-section profiles were evaluated, i.e., rectangular, triangular and trapezoidal. They found that rectangular cross-section offers the best performance. In addition, it was highlighted that low pressure resistance and high pressure drop were observed for micro-channels which have a high aspect ratio, long wetted perimeter and small hydraulic diameter.

Yang et al. [3] numerically and experimentally evaluated cooling performance of a micro-channel heat sink with five different pin-fin configurations, i.e., circle, hexagon, pentagon, square and triangle. The numerical results revealed that hexagon pin fin offers the lowest thermal resistance with better uniformity while the circular pin fin induce the lowest pressure drop. A similar study was reported by Zhao et al. [4], who conducted optimization of the heat transfer performance of micro-channel heat sinks with square pin fins. Both pin-fin porosity and located angle were evaluated and the result indicated a strong dependency of the cooling performance on both parameters and an optimum value of these parameters offer higher cooling performance. Al Neama et al. [5] proposed and evaluated heat transfer performance of a serpentine cooling channel equipped with a chevron fin structure. The proposed design was found to offer better cooling performance, mirrored by lower thermal resistance and lower pressure drop. A comprehensive review on the heat transfer performance of a micro-channel heat sink was provided by Ghani et al. [6]

Recently, rectilinear channel designs with oblique fins have attracted considerable attention due to their high cooling capability and low pumping power requirement. Lee et al. [7] conducted a parametric study to investigate the fluid flow and heat transfer performance of a micro-channel heat sink with oblique fins. They concluded that oblique fins induce re-initialization of the boundary layer and secondary flow yield for better heat transfer performance with no significant changes in pressure drop. Their study also indicated that better performance can be achieved by having smaller oblique angles and smaller fin pitches. Mou et al. [8] adopted an equivalent circuit model to study the mass flow and temperature distribution in oblique fin heat exchangers. The developed model offers the capability to predict the mass flow and temperature distribution for different parameters with good accuracy. Meanwhile, the effect of cross-section profile in a cooling channel with oblique fins was investigated by Vinoth and Kumar [9]. Three different cross-section profiles were studied, namely semicircle, square and trapezoidal. It was found that trapezoidal cross-section yields the best heat transfer performance, albeit it also imposes the highest pressure drop.

The transient heat transfer performance of a parallel channel with oblique fins was evaluated and reported by Prajapati et al. [10]. By conducting experimental investigations, they measured and compared the performance of a traditional parallel channel with that equipped with oblique fins. The latter was found to have better heat transfer and shorter response time as compared to the former. Ghani et al. [11] proposed heat transfer enhancement of a micro-channel heat sink by utilising a hybrid technique of ribs and secondary channels which is closely similar to the oblique fin configuration. They found that a combination of both ribs and secondary channels offered better performance as compared to the case where they were implemented individually. Om et al. [12] evaluated the effect of the oblique fin arrangement on the flow behaviour and heat transfer performance of a liquid cooling plate. They investigated inline, inclined and louvered configurations and found that the last configuration offered the best cooling performance.

Most of these studies have been focused on the performance evaluation based on the first thermodynamic principle. None of them consider entropy generation when evaluating the performance of the considered cooling channel designs. Meanwhile, some studies have highlighted the importance of second law thermodynamics in the performance evaluation of a thermal system [13], e.g., cooling channels with porous plates [14], a regenerative cooling channel [15], chemical reactors [16], jet impingement cooling [17], nanofluid flow in a square cavity [18], and cooling of a data center room [19]. By utilizing entropy generation analysis in combination with computational fluid dynamics approach, local losses due to heat transfer, friction, turbulent, mass transfer and phase change can be quantified [20]. In our previous studies [21,22,23], we have adopted this approach to evaluate the heat transfer performance of a helical coil tube subjected to a large temperature difference. In this study, the same approach will be adopted for the performance evaluation of a rectilinear cooling channel design with oblique fins. The main objective of this study is to evaluate the performance of parallel with oblique channels with regards to the first and second thermodynamic point of view. The effect of oblique fin angle will be evaluated to obtain the optimum conditions. Several inlet Reynolds numbers and wall heat fluxes will be evaluated to provide a comprehensive overview of the cooling channel performance.

## 2. Mathematical Model

In this study, a conjugate heat transfer between a solid separator where the cooling channel is engraved and a cooling liquid in considered. A schematic of the considered system is presented in Figure 1. At the base of the solid separator, a constant heat flux is applied to represent heat dissipated by an electronic component. Isotropic thermal conductivity is assumed for the solid separator while the liquid is assumed to be an incompressible Newtonian fluid. The studied flow is laminar and the area of the electronic component that generates heat is kept constant to ensure consistent performance evaluation for all channel designs. Details of the geometric parameters are listed in Table 1.

### 2.1. Governing Equations

Only conduction heat transfer is considered for the solid separator. Hence, the conservation of energy for the solid separator is given by:(1)$$k_{s}\nabla^{2}T=0$$
where *k_s_* is the conductivity of the solid separator and *T* is temperature.

Meanwhile, simultaneous fluid flow and convective heat transfer is considered for the cooling channel. The conservation of mass, momentum and energy for this channel are therefore expressed as:(2)$$\nabla \cdot \rho_{w}u=0$$
(3)$$\nabla \cdot \left(\rho_{w}uu\right)=-\nabla pI+\nabla \cdot \left[\mu_{w}\left(\nabla u+{\left(\nabla u\right)}^{T}\right)\right]+\rho_{w}g$$
(4)$$\rho_{w}c_{p,w}u\cdot \nabla T=\nabla \cdot \left(k_{w}\nabla T\right)$$
where *ρ_w_* represents fluid density, **u** is the fluid velocity, *p* is the pressure, **I** is identity tensor, *μ_w_* is the fluid dynamic viscosity, **g** is gravity acceleration, *c_p,w_* is the fluid specific heat, *k_w_* is fluid thermal conductivity and *T* is temperature.

Entropy balance equation for an open system [13,21] is adopted to take into account the entropy generation inside the cooling channel. This balance equation is given by:(5)$$-\nabla \sigma +s_{g}=0$$
where **σ** is the entropy flux and *s_g_* is the entropy generation rate per unit volume.

### 2.2. Constitutive Relations

The entropy generation, *s_g_*, in Equation (5) has four contributing components, i.e., heat transfer contributions, viscous dissipation contribution, mass transfer contribution and chemical reaction contribution [13]. In this study, however, no mass transfer and chemical reaction occurs, thus only the first two are considered, for which:(6)$$s_{g}=s_{h}+s_{\mu }$$
where, the heat transfer contribution *s_h_* and viscous dissipation contribution *s_μ_* are given by [13]:(7)$$s_{h}=\frac{\nabla \cdot \left(k\nabla T\right)}{T^{2}}$$
(8)$$s_{\mu }=-\frac{1}{T}\tau :\nabla u$$

In the above equation, the viscous stress tensor **τ** is given by second term right hand side of Equation (3) in this study. Similar to our previous study [1], to evaluate the cooling channel performance, figure of merit (FoM) parameter is adopted and is defined as ratio of the heat transfer rate to the pumping power, i.e.:(9)$$FoM=\frac{\dot{Q}}{P_{pump}}$$

The total heat transfer rate and pumping power can be expressed as:(10)$$P_{pump}=\frac{\Delta p\dot{V}}{\eta_{pump}}$$
(11)$$\dot{Q}=\int_{A_{t}}\dot{q}dA_{t}$$
respectively. In the above equation $\dot{V}$ is the cooling liquid volumetric flow rate, Δ*p* is the pressure difference between the channel inlet and outlet, $\dot{q}$ is the heat flux from the electronic chips and *A_t_* is the heat transfer area (chip top surface area). In addition to the maximum temperature, the standard deviation of the temperature of the heat transfer area will be compared to evaluate the uniformity of temperature distribution for each channel design, i.e.:(12)$$\sigma_{std}={\left(\frac{1}{A_{t}}\int_{A_{t}}{\left(T-T_{ave}\right)}^{2}dA_{t}\right)}^{1/2}$$

The average temperature, *T_ave_* is given by:(13)$$T_{ave}=\int_{A_{t}}TdA_{t}$$

For the performance evaluation with regards to the second thermodynamics law, the global entropy generation will be presented and discussed. This parameter can be obtained by integration of the entropy generation rate per unit volume over the entire cooling channel volume (V), i.e.:(14)S˙g=∫VsgdV

To evaluate the significance of each entropy generation component, the Bejan number is utilized and is defined as:(15)$$\mathrm{Be}=\frac{s_{h}}{s_{g}}$$

### 2.3. Boundary Condition

To complete the developed model, the following boundary conditions are imposed:

Inlet: Constant inlet mass flow rate and temperature are set:(16)$$\dot{m}={\dot{m}}_{in}=\rho A_{c}U_{in},\;T=T_{in}$$
where *A_c_* is the inlet cross-section area of the cooling channel, *U_in_* is the inlet velocity.

Outlet: Gauge pressure and stream-wise temperature gradient are set to zero: (17)$$p=p_{out},\;n\cdot \nabla T=0$$

Bottom surface of the solid separator: A constant heat flux is set, representing the heat dissipated from the electronic chips:(18)$$n\cdot \left(k_{s}\nabla T\right)={\dot{q}}_{base}$$

At the interface between cooling liquid and solid: A no slip condition and coupled temperature are prescribed:(19)$$u=0,{T_{s}|}_{\mathrm{int}}={T_{w}|}_{\mathrm{int}}$$

Side wall: Adiabatic condition is set:(20)$$n\cdot \left(k_{s}\nabla T\right)=0$$

Top wall (cooling liquid part): No slip condition with zero heat flux is specified:(21)$$u=0,\;n\cdot \left(k_{w}\nabla T\right)=0$$

Top wall (solid channel separator part): Zero heat flux is specified:(22)$$n\cdot \left(k_{s}\nabla T\right)=0$$

The mass flow rate studied in this study corresponds to inlet Reynolds numbers of 100, 250, 500, 750 and 1000, while two heat flux values are chosen, i.e., 10,000 W/m^2^, which represents low heat density electronic chips/fuel cell and 500,000 W/m^2^, which represents the current heat dissipation from high performance microprocessor chips. It should be noted that the boundary conditions for entropy balance equation are similar to those for conservation of momentum and energy.

### 2.4. Numerical Methodology

The computational domain, consisting of solid separator and liquid cooling channel, were created using the ANSYS Design modeller and meshed in ANSYS Meshing. Several mesh sizes were prepared to study the dependency of the numerical result on the amount of mesh. After the boundary conditions have been completely labelled, the computational domain was exported to ANSYS Fluent for model set-up. The conservation equations together with constitutive relations and corresponding boundary conditions were solved by using the widely adopted Semi-Implicit-Pressure-Linked equation (SIMPLE) algorithm, second order upwind discretization and algebraic multi-grid (AMG) method. A residual criterion of 10 is set for all parameters. The computational model takes approximately 30 min to converge using single processor setting in high performance computer (HPC). A range of 2 GB to 4 GB RAM utilization was recorded during computational run where other processes are closed.

To evaluate the mesh independency of the numerical result, a mesh-independent study was conducted using the previously prepared mesh. The results are presented in Figure 2. As can be seen, no significant changes on outlet temperature are observed after meshes of 2.5 million. Consequently, a mesh size amounting 2.5 million meshes was chosen for all cases.

## 3. Results and Discussion

### 3.1. Model Validation

To validate the developed model, the model prediction of the channel outlet temperature is compared to the analytical solution. The comparison is presented in Table 2. As can be seen, a good agreement is achieved between the present model prediction and the analytical solution with less than 4% relative error for all studied *Re* ranges. This finding firmly indicates the validity of the developed model.

### 3.2. Effect of Channel Geometry

The study compares six different channels geometries, emphasizing the importance of microchannel geometry on the cooling performance. The variations of velocity at the middle of the channel (*z* = 5 × 10^−4^ m) for various channel geometries are presented in Figure 3. It can be seen that velocity in the center zone of the cooling channel is significantly lower than at the inlet. This is due to more fluid flowing through the inlet and outlet manifolds since they have lesser resistance. Therefore, a more uniform distribution can be achieved by employing, for instance, fractal channels. Moreover, Figure 3 shows that in all channels, a relatively uniform velocity is observed throughout the middle zone. 

The channel geometry also greatly affects the heat transfer performance and hence, the temperature profile of both the fluid and the chip surface. The temperature distributions are depicted in Figure 4 and Figure 5, respectively. From Figure 4, it is clear that the cooling fluid temperature in the middle zone of the parallel channel is much higher, compared to that of oblique channels under the same conditions (at *Re* 1000 and *Q_base_* 10,000 W/m^2^), while all oblique fins with different angles show nearly uniform temperature profiles. For the same conditions, the temperature distributions at the base of the solid separator (*z* = −1 × 10^−3^ m), which represents the surface of the electronic chip, were investigated. It can be inferred from Figure 5 that the maximum temperature of the base with the parallel channel is significantly higher than that of oblique fin channel, which is observed near the outlet region.

It is also important to look at the second law thermodynamic analysis of the different channel geometries. Figure 6 shows contour of the entropy generation at the chip surface for all channels for the same flow manner. From this figure, it can be seen that, entropy generation is higher for the conventional parallel channel than the oblique fin channels. Moreover, higher entropy in a parallel channel is observed near the inlet and close to the outlet region, with a magnitude about three times higher than in the oblique fin channels.

### 3.3. Effect of Mass Flow Rate

Another important factor that has an effect on the heat transfer is the mass flow rate of the coolant. The maximum temperature, average temperature and standard deviation of temperature at the solid separator base, as well as the pressure drop across the channels, as affected by different mass flow rates, which correspond to various Reynolds numbers, are shown in Figure 7. 

From Figure 7a,b, it can be observed that channels with an oblique angle of 90° have lower maximum and average base temperature (~3 to 5 °C), indicating better heat dissipation. At low *Re*, a lower standard deviation value is observed for the parallel channel, as shown in Figure 7c, suggesting that parallel channel has better uniformity. However, the observed behaviour changes at high *Re*, where standard deviation of the base temperature is lowest for oblique 90° fins due to the secondary flow that is able to remove more heat and break the fully developed flow to create a new boundary layer and, thus enhance the heat transfer. All channels show the same general trend in terms of pressure drop-increasing with Reynolds number (Figure 7d). Overall, the parallel channels impose higher pressure drops, while oblique channels with an angle of 20° have the lowest pressure drop.

### 3.4. Effect of Base Heat Flux

Constant base heat flux has been employed for the purpose of the previous analysis. It is also of interest to see how heat flux can affect the chip surface temperature and flow behaviour due to temperature-dependent thermophysical properties of the fluid. Intuitively, the temperature is expected to rise as the heat flux is raised, as shown in Figure 8a,b. Higher heat flux imposes a higher maximum and average temperature on the solid separator base. It is noted that at lower heat flux, the temperature difference between parallel and oblique fin channels is about 2 °C, while at higher heat flux, this temperature difference raises to 15 °C, respectively. The parallel channel design gives the highest average temperature and shows a steeper positive slope, suggesting it is relatively more sensitive to the base heat flux than the oblique channel design. In terms of uniformity (Figure 8c), the temperature uniformity in all channel designs is also found to follow the same trend; proportionally increasing with the prescribed heat flux. This increasing standard deviation indicates that the temperature distribution becomes less uniform as the base heat flux increases. The pressure drop required for a parallel channel is about 25% more than that of an oblique fin channel (Figure 8d).

### 3.5. Overall Heat Transfer Performance

A further point of interest in this study is the overall heat transfer performance of the channels. Table 3 summarizes the Figure of Merit (FoM) defined in Equation (9). Here, several features are apparent; foremost among them is that the FoM decreases as the *Re* is increased. Notably, the FoM is about two orders-of-magnitude lower as the *Re* is increased from 100 to 1000, which can be explained by the exponential increase in pressure drop (Figure 7d). The oblique fin channel gives rise to a higher FoM as compared to a parallel channel. On closer inspection, the oblique fin with 20° angle yields the highest FoM, especially at higher *Re*, which shows its potential application to balance the heat transfer performance and pumping requirement at higher *Re*. At lower *Re*, on the other hand, the oblique fin with 60° angle shows the best FoM. With regard to heat flux from the chip, in general, increasing the heat flux increases the FoM due to higher convective flux. In all cases, the oblique fin channel with 20° angle yields the best FoM.

Now, looking at the total entropy generation in the system as summarized in Table 4, it is seen that when the *Re* is increased from 100 to 1000, the total entropy generation decreases and increases by one order-of-magnitude for heat transfer and friction entropy generation, respectively. Closer inspection reveals that the entropy generation due to viscous dissipation is about three orders-of-magnitude lower than that of heat transfer entropy. Interestingly, at low *Re* number, the parallel design yields the lowest and the highest entropy generation due to heat transfer at the chip and liquid channel, respectively. At higher *Re* number, on the contrary, the oblique-fin design with 90° angle gives the lowest entropy generation (about half that of the parallel counterpart).

Table 5 summarizes the total entropy generation at different heat fluxes. Essentially, a higher heat flux generates higher entropy generation, especially for the heat transfer. Notably, the entropy generation due to heat transfer increases by about 20–25 times when the heat flux is increased by five times. Meanwhile, the entropy generation due to viscous dissipation decreases by about 3–5% when the heat flux is increased from 10,000 to 50,000 w/m^2^. Amongst all the channels, the oblique fin channel with 90° angle generates the lowest entropy generation in the chip, whereas the oblique fin channel with 20° angle produces the lowest entropy generation in the cooling liquid.

## 4. Conclusions

Heat transfer performance and entropy generation in a cooling channel with oblique fins have been numerically investigated and discussed. The effect of oblique fin angle was evaluated by comparing the heat transfer performance and entropy generation. The performance of a parallel cooling channel was taken as a benchmark for this comparison and evaluation. The results indicate that oblique fin channel has higher heat transfer performance, lower pressure drop, lower entropy generation as well as better figure of merit (FoM). In addition, the effect of inlet Reynolds number was examined by varying the inlet mass flow rate. It was found that increasing *Re* number decreases the FoM and increasing heat flux increases the FoM. In terms of entropy generation, it can be concluded that entropy generation due to heat transfer is about three orders-of-magnitude higher than the entropy generation due to viscous dissipation. Increasing Reynolds number decreases the entropy generation and oblique fins with 90° angle generate the lowest entropy generation.

## Figures and Tables

**Figure 1 entropy-21-00191-f001:**
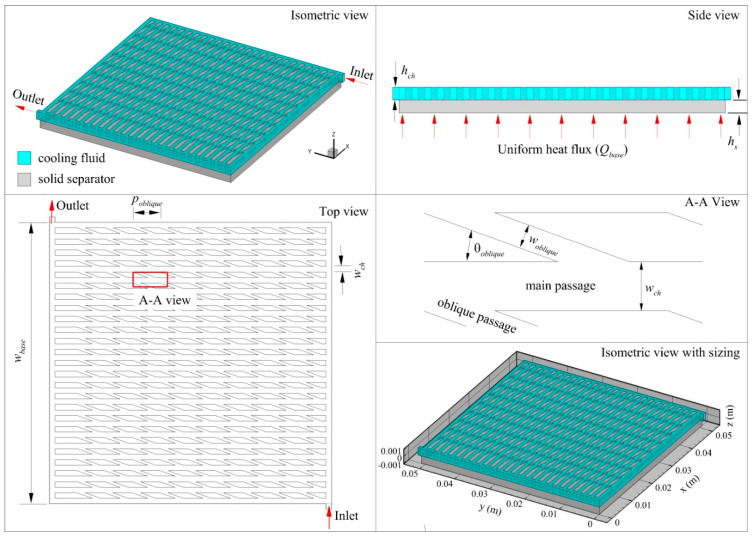
Schematics of the investigated cooling channel.

**Figure 2 entropy-21-00191-f002:**
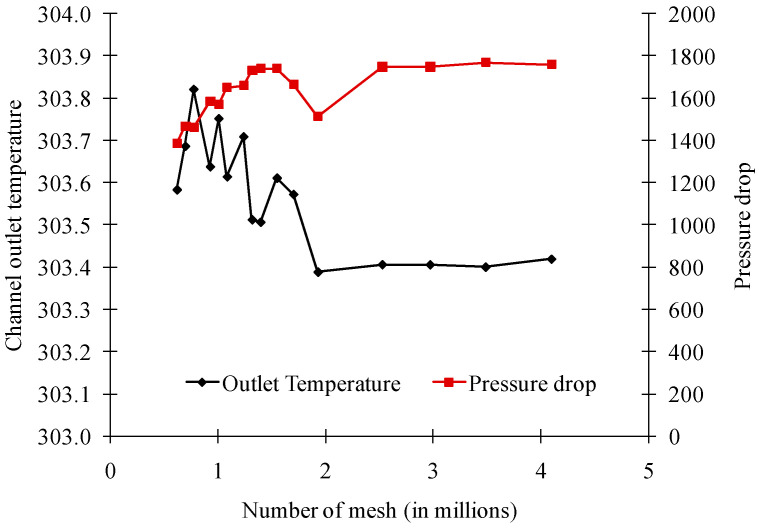
Cooling channel outlet temperature and pressure drop for various mesh sizes.

**Figure 3 entropy-21-00191-f003:**
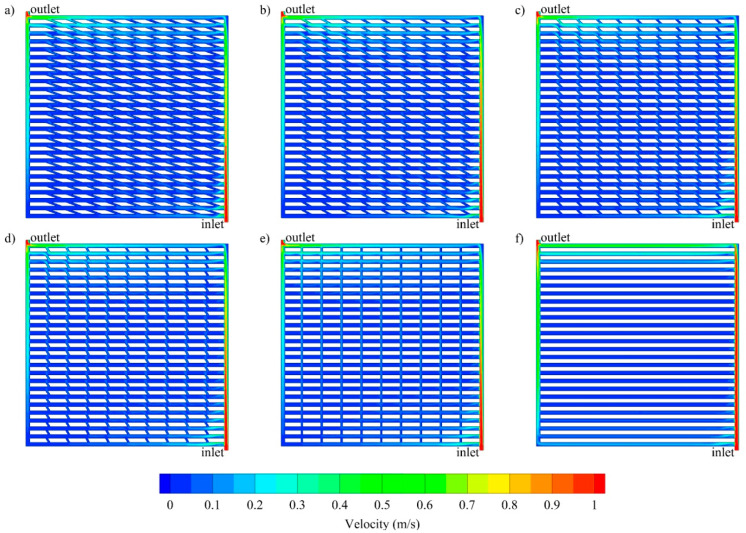
Velocity contours at the middle of cooling channel (*z* = 5 × 10^−4^ m) for oblique channels with various oblique angle: (**a**) 20°, (**b**) 30°, (**c**) 45°, (**d**) 60°, (**e**) 90° and (**f**) parallel channel.

**Figure 4 entropy-21-00191-f004:**
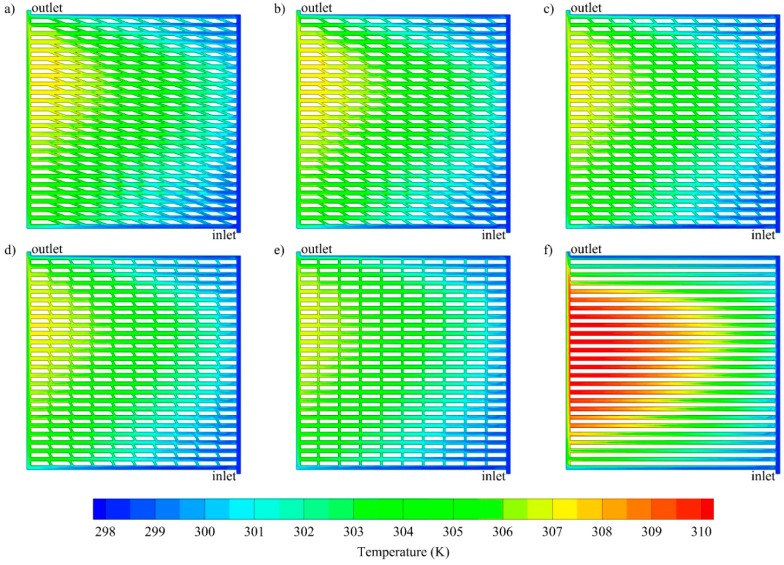
Temperature distribution at the middle of cooling channel (z = 5 × 10^−4^ m) for oblique channels with various oblique angle: (**a**) 20°, (**b**) 30°, (**c**) 45°, (**d**) 60°, (**e**) 90° and (**f**) parallel channel at *Re* 1000 and *Q_base_* 10,000 W/m^2^.

**Figure 5 entropy-21-00191-f005:**
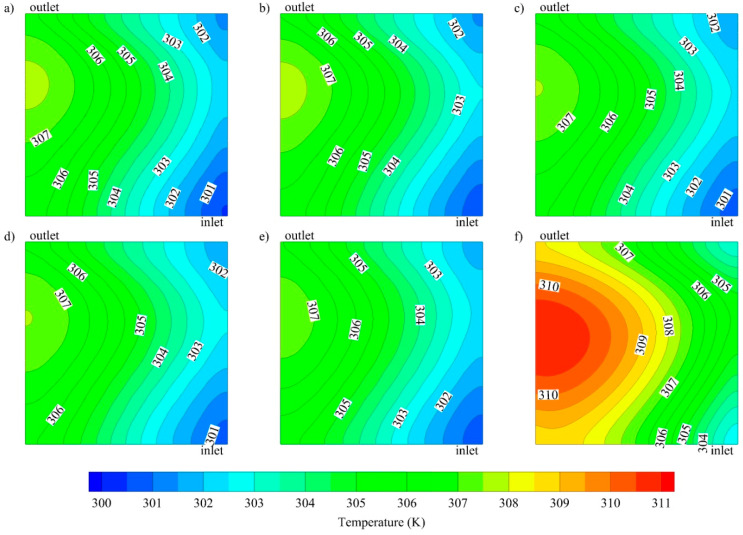
Temperature distribution at the base of the solid separator (*z* = −1 × 10^−3^ m) for oblique channels with various oblique angle: (**a**) 20°, (**b**) 30°, (**c**) 45°, (**d**) 60°, (**e**) 90° and (**f**) parallel channelat *Re* 1000 and *Q_base_* 10,000 W/m^2^.

**Figure 6 entropy-21-00191-f006:**
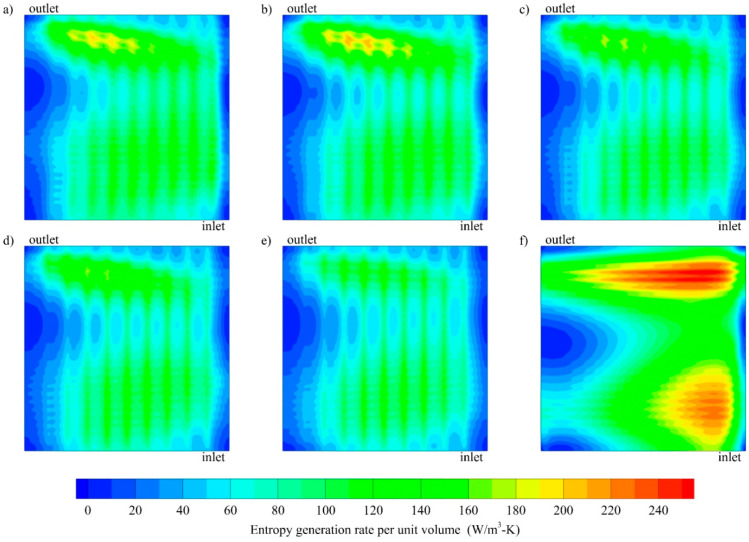
Contour of entropy generation at the base of the solid separator (*z* = −1 × 10^−3^ m) for oblique channels with various oblique angle: (**a**) 20°, (**b**) 30°, (**c**) 45°, (**d**) 60°, (**e**) 90° and (**f**) parallel channel at *Re* 1000 and *Q_base_* 10,000 W/m^2^.

**Figure 7 entropy-21-00191-f007:**
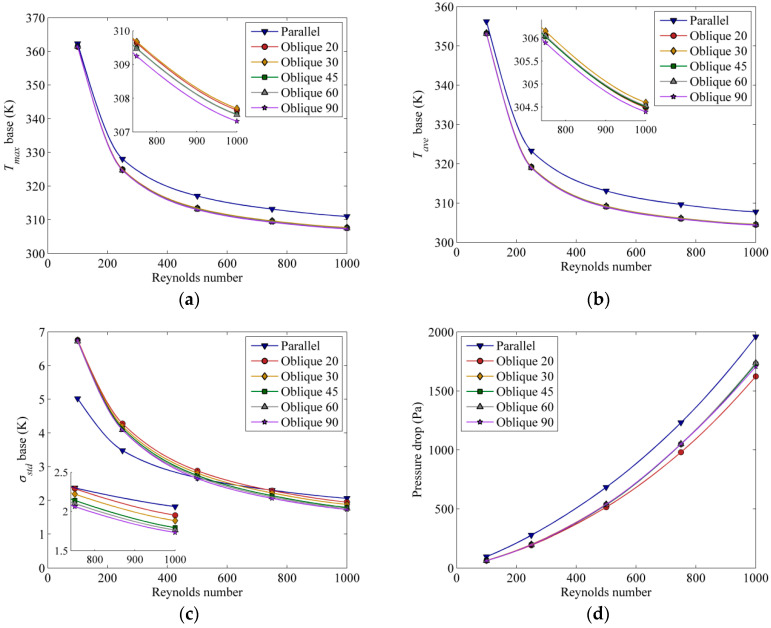
(**a**) Maximum temperature, (**b**) average temperature, and (**c**) standard deviation of temperature at the base of the solid separator (*z* = −1 × 10^−3^ m) and (**d**) cooling channel pressure drop for various inlet Reynolds number at constant base heat flux of 10,000 W/m^2^.

**Figure 8 entropy-21-00191-f008:**
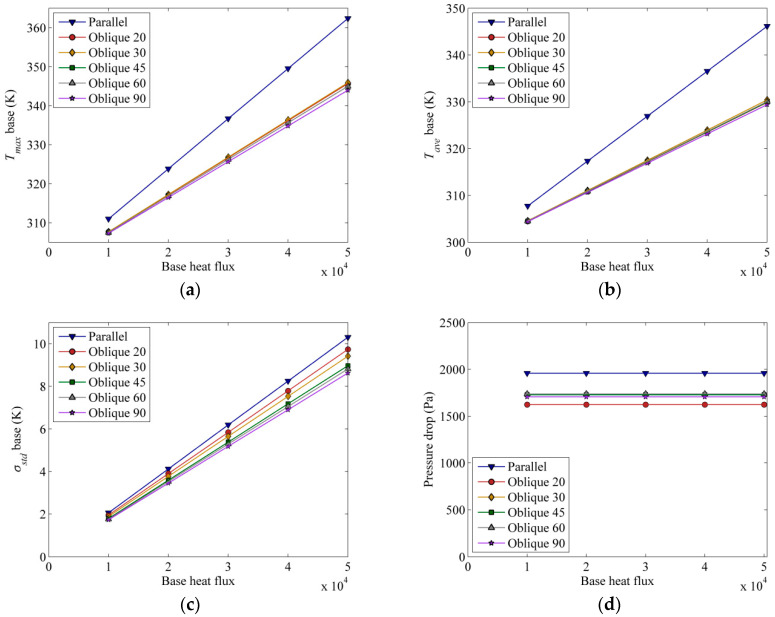
(**a**) Maximum temperature, (**b**) average temperature, and (**c**) standard deviation oftemperature at the base of the solid separator (*z* = −1 × 10^−3^ m) and (**d**) cooling channel pressure drop for various constant base heat flux at inlet Reynolds number of 1000 of 10,000 W/m^2^ and constant inlet Reynolds number of 1000.

**Table 1 entropy-21-00191-t001:** Parameters, operating parameters and material properties.

Parameter	Symbol	Value	Unit
Base (heated wall) width	*w_base_*	5.10 × 10^−2^	m
Channel width	*w_ch_*	1.00 × 10^−3^	m
Channel height	*h_ch_*	1.00 × 10^−3^	m
Separator height	*h_s_*	1.00 × 10^−3^	m
Oblique fin angle	*θ_oblique_*	20, 30, 45, 60 and 90	degree
Oblique fin width	*w_oblique_*	5.00 × 10^−4^	m
Oblique fin pitch	*p_oblique_*	5.00 × 10^−2^	m
Total length of the channel	*L_ch_*	1.376	m
Inlet mass flow rate	${\dot{m}}_{in}$	1.00 × 10^−4^ (*Re* 100), 2.50 × 10^−4^ (*Re* 250), 5.00 × 10^−4^ (*Re* 500), 7.50 × 10^−4^ (*Re* 750), 1.00 × 10^−3^ (*Re* 1000)	kg/s
Outlet Pressure	*P_out_*	101,325 (1 atm)	Pa
Inlet temperature	*T_in_*	298.15	K
Thermal conductivity of solid separator	*k_s_*	387.6	W/m^2^·K
Density of cooling fluid	*ρ_w_*	998.2	kg/m^3^
Viscosity of cooling fluid	*μ_w_*	10.03 × 10^−3^	Pa·s
Thermal conductivity of cooling fluid	*k_w_*	0.6	W/m^2^·K
Specific heat capacity of cooling fluid	*c_p,w_*	4182	W/kg·K
Solid separator base heat flux	${\dot{q}}_{base}$	10,000 (base case), 20,000, 30,000, 40,000 and 50,000	W/m^2^

**Table 2 entropy-21-00191-t002:** Cooling channel outlet temperature (K).

Inlet Reynolds Number	100	250	500	750	1000
Analytical solution (K)	360.16	322.95	310.55	307.01	304.35
Present simulation (K)	359.90	322.24	310.20	305.98	303.78
Deviation (K)	0.26	0.71	0.35	1.03	0.57

**Table 3 entropy-21-00191-t003:** Figure of merit (in thousands) for various configurations.

**Geometry**	**Inlet Reynolds Number**				
	**100**	**250**	**500**	**750**	**1000**
Parallel	1898.28	261.63	53.09	21.03	9.25
Oblique 20	2873.38	376.02	70.49	26.41	11.16
Oblique 30	2828.31	362.70	67.31	24.67	10.44
Oblique 45	2907.87	369.46	67.96	24.71	10.49
Oblique 60	2934.62	369.57	67.50	24.63	10.43
Oblique 90	2924.93	368.80	67.47	24.69	10.62
	**Base Heat flux (W/m^2^)**				
	**10,000**	**20,000**	**30,000**	**40,000**	**50,000**
Parallel	9.25	18.50	27.75	37.00	46.48
Oblique 20	11.16	22.33	33.49	44.65	56.08
Oblique 30	10.44	20.88	31.33	41.77	52.46
Oblique 45	10.49	20.97	31.46	41.94	52.68
Oblique 60	10.43	20.87	31.30	41.73	52.42
Oblique 90	10.62	21.24	31.86	42.48	53.36

**Table 4 entropy-21-00191-t004:** Global entropy generation for various inlet Reynold number at constant base heat flux of 10,000 W/m^2^.

*Re*	Geometry	Global Entropy Generation					Liquid Bejan Number
		Heat Transfer (×10^−4^ W/k)		Viscous Dissipation (×10^−7^ W/k)		Total (×10^−4^ W/k)	
		Solid	Liquid	Solid	Liquid		
100	Parallel	16.00	44.82	0.00	0.28	60.83	1.00
	Oblique 20	21.79	33.59	0.00	0.18	55.37	1.00
	Oblique 30	21.84	33.70	0.00	0.18	55.54	1.00
	Oblique 45	21.97	33.06	0.00	0.17	55.04	1.00
	Oblique 60	21.89	33.45	0.00	0.17	55.34	1.00
	Oblique 90	21.90	33.21	0.00	0.17	55.11	1.00
250	Parallel	9.42	21.50	0.00	2.00	30.92	1.00
	Oblique 20	9.94	12.38	0.00	1.31	22.32	1.00
	Oblique 30	9.65	12.62	0.00	1.37	22.27	1.00
	Oblique 45	9.41	12.38	0.00	1.32	21.80	1.00
	Oblique 60	9.27	12.55	0.00	1.30	21.82	1.00
	Oblique 90	9.11	12.44	0.00	1.30	21.55	1.00
500	Parallel	6.21	14.36	0.00	9.14	20.57	1.00
	Oblique 20	4.98	7.20	0.00	6.13	12.18	1.00
	Oblique 30	4.85	7.52	0.00	6.51	12.38	1.00
	Oblique 45	4.58	7.35	0.00	6.22	11.93	1.00
	Oblique 60	4.49	7.48	0.00	6.18	11.97	1.00
	Oblique 90	4.22	7.34	0.00	6.13	11.57	1.00
750	Parallel	4.87	11.78	0.00	22.51	16.68	1.00
	Oblique 20	3.39	5.64	0.00	15.36	9.04	1.00
	Oblique 30	3.32	5.99	0.00	16.49	9.33	1.00
	Oblique 45	3.05	5.88	0.00	15.62	8.95	1.00
	Oblique 60	2.99	5.99	0.00	15.57	9.00	1.00
	Oblique 90	2.77	5.85	0.00	15.40	8.64	1.00
1000	Parallel	4.07	10.30	0.00	42.71	14.41	1.00
	Oblique 20	2.58	4.84	0.00	29.73	7.46	0.99
	Oblique 30	2.55	5.15	0.00	32.08	7.73	0.99
	Oblique 45	2.31	5.09	0.00	30.19	7.43	0.99
	Oblique 60	2.26	5.19	0.00	30.21	7.48	0.99
	Oblique 90	2.08	5.06	0.00	29.93	7.18	0.99

**Table 5 entropy-21-00191-t005:** Global entropy generation for various applied heat flux at constant inlet *Re* of 1000.

Base Heat Flux (W/m^2^)	Geometry	Global Entropy Generation (×10^−4^ W/k)					Liquid Bejan Number
		Heat Transfer (×10^−4^ W/k)		Viscous Dissipation (×10^−7^ W/k)		Total (×10^−4^ W/k)	
		Solid	Liquid	Solid	Liquid		
10,000	Parallel	4.07	10.30	0.00	42.71	14.41	1.00
	Oblique 20	2.58	4.84	0.00	29.73	7.46	0.99
	Oblique 30	2.55	5.15	0.00	32.08	7.73	0.99
	Oblique 45	2.31	5.09	0.00	30.19	7.43	0.99
	Oblique 60	2.26	5.19	0.00	30.21	7.48	0.99
	Oblique 90	2.08	5.06	0.00	29.93	7.18	0.99
20,000	Parallel	15.40	39.82	0.00	42.32	55.26	1.00
	Oblique 20	9.95	18.93	0.00	29.50	28.90	1.00
	Oblique 30	9.80	20.13	0.00	31.82	29.97	1.00
	Oblique 45	8.88	19.92	0.00	29.95	28.83	1.00
	Oblique 60	8.69	20.31	0.00	29.96	29.03	1.00
	Oblique 90	8.02	19.81	0.00	29.67	27.85	1.00
30,000	Parallel	32.81	86.68	0.00	41.94	119.53	1.00
	Oblique 20	21.54	41.64	0.00	29.27	63.21	1.00
	Oblique 30	21.21	44.31	0.00	31.58	65.55	1.00
	Oblique 45	19.22	43.85	0.00	29.72	63.10	1.00
	Oblique 60	18.81	44.71	0.00	29.72	63.54	1.00
	Oblique 90	17.36	43.60	0.00	29.41	60.99	1.00
40,000	Parallel	55.31	149.23	0.00	41.58	204.58	1.00
	Oblique 20	36.90	72.41	0.00	29.05	109.34	1.00
	Oblique 30	36.29	77.07	0.00	31.34	113.40	1.00
	Oblique 45	32.89	76.30	0.00	29.49	109.22	1.00
	Oblique 60	32.18	77.77	0.00	29.49	109.99	1.00
	Oblique 90	29.74	75.84	0.00	29.17	105.61	1.00
50,000	Parallel	82.08	226.00	0.00	41.24	308.12	1.00
	Oblique 20	55.60	110.72	0.00	28.84	166.34	1.00
	Oblique 30	54.63	117.88	0.00	31.11	172.55	1.00
	Oblique 45	49.53	116.73	0.00	29.28	166.28	1.00
	Oblique 60	48.45	118.97	0.00	29.26	167.44	1.00
	Oblique 90	44.80	116.02	0.00	28.94	160.85	1.00

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
