# Peer review of "Entropy Generation and Heat Transfer Performance in Microchannel Cooling"

_entropy, 2019, doi:10.3390/e21020191_

Reviewer 1 Report

The study shows interesting aspects, and from my point of view it can be accepted in Entropy after some observations:

Check line 83.

Carefully review the nomenclature throughout the document, some of them do not match the format, example, sg.

Please improve figures 7 and 8, it is difficult to appreciate the different angles. Maybe using colors in different behaviors.

Author Response

We are very grateful for the positive reviews, comments, suggestions and the time spent reviewing our paper (Ref. No.: entropy-441815). In the following, we have addressed all the issues raised by the reviewer in the revised paper. Below, we list the reviewer’s comments followed by our responses. The changes are highlighted by using track changes in the revised manuscript:

·       The study shows interesting aspects, and from my point of view it can be accepted in Entropy after some observations

Thank you for your very encouraging comment.

·       Check line 83.

The sentence has been checked and revised accordingly.

·       Carefully review the nomenclature throughout the document, some of them do not match the format, example, sg.

The nomenclature has been thoroughly reviewed as suggested.

·       Please improve figures 7 and 8, it is difficult to appreciate the different angles. Maybe using colors in different behaviors.

We have tried to implement different color as suggested. However, the difference is still not significant. We therefore change the figure with line and symbol as shown.

Reviewer 2 Report

In this paper, the authors investigate the flow and heat transfer of an incompressible fluid in a cooling channel with oblique fins. The results may be helpful for designing the cooling system of electronic devices. Therefore, i recommend the publication of this paper in the Entropy after the following questions are addressed:

(1) There are some grammatical errors in the manuscript. For example, the sentences in line 83 and 93 on page 2, line 200 on page 7. "Eq.3" in line 138 on page 5 should be changed to "Eq. (3)".

(2) A whole table should be put in one page. 

(3) This study indicates that the different evaluation criterion of heat transfer enhancement leads to a different optimization result. The question is which optimization result should be adopted in the practical desgin of the cooling system of the electronic devices?

Author Response

Response to Reviewer #2

We are very grateful for the positive reviews, comments, suggestions and the time spent reviewing our paper (Ref. No.: entropy-441815). In the following, we have addressed all the issues raised by the reviewer in the revised paper. Below, we list the reviewer’s comments followed by our responses. The changes are highlighted by using track changes in the revised manuscript:

In this paper, the authors investigate the flow and heat transfer of an incompressible fluid in a cooling channel with oblique fins. The results may be helpful for designing the cooling system of electronic devices. Therefore, I recommend the publication of this paper in the Entropy after the following questions are addressed:

Thank you for your very encouraging comment.

·       There are some grammatical errors in the manuscript. For example, the sentences in line 83 and 93 on page 2, line 200 on page 7. "Eq.3" in line 138 on page 5 should be changed to "Eq. (3)".

Thank you for pointing out these errors, we have revised these accordingly. In addition, the whole manuscript has been thoroughly proofread to minimize such error.

·       A whole table should be put in one page.

Indeed, we agree with the reviewer comments. Nevertheless, final typesetting and formatting will be handled by the journal editorial team.

·       This study indicates that the different evaluation criterion of heat transfer enhancement leads to a different optimization result. The question is which optimization result should be adopted in the practical design of the cooling system of the electronic devices?.

Thank you for pointing out this issue. Based on the obtained result, channels with oblique angle of 90° is the most superior configuration for the cooling purposes: albeit it has high pressure drop, it offers more uniform temperature distribution and lower maximum temperature which is crucial in cooling application. In addition, it generates the lowest entropy which is good from the second thermodynamic law perspective.

Reviewer 3 Report

Cooling technology has been a major concern for electronic devices and up to now, several approaches have been proposed to improve its performance. The current work investigates heat transfer performance and entropy generation in a cooling channel with oblique fins mainly through numerical methods. Generally speaking, although the topic is interesting, some fetal problems exist. I do not recommend its publication in its current form.

1. What is the relationship between heat transfer and entropy generation for evaluating performance of microchannel cooling?

2. Recently, several literatures has been published for evaluating heat transfer and fluid flow characteristics for microchannel cooling, and it is suggested that the author could review more works.

3. In the simulation, cases for oblique angle 20, 30, 45, 60 and 90 were tested. It is suggested that the author could give the reason why those cases were selected. Besides, in an actual situation, which oblique angle will be optimal choice?

4. The current work was performed mainly through numerical methods, and how the author validate the results corresponding actual application? 

Author Response

Response to Reviewer #3

We are very grateful for the positive reviews, comments, suggestions and the time spent reviewing our paper (Ref. No.: entropy-441815). In the following, we have addressed all the issues raised by the reviewer in the revised paper. Below, we list the reviewer’s comments followed by our responses. The changes are highlighted by using track changes in the revised manuscript:

Cooling technology has been a major concern for electronic devices and up to now, several approaches have been proposed to improve its performance. The current work investigates heat transfer performance and entropy generation in a cooling channel with oblique fins mainly through numerical methods. Generally speaking, although the topic is interesting, some fetal problems exist. I do not recommend its publication in its current form.

·       What is the relationship between heat transfer and entropy generation for evaluating performance of microchannel cooling?

Thank you for your valuable comment. As stated in the introduction, entropy generation analysis has become important tools in determining the overall performance of a thermal system. Several studies have highlighted this finding and they have been cited in the manuscript.

·       Recently, several literatures has been published for evaluating heat transfer and fluid flow characteristics for microchannel cooling, and it is suggested that the author could review more works..

Indeed, we have extensively reviewed recent literatures in cooling channel design as can be found in the manuscript. However, none of them considered entropy generation analysis coupled with computational fluid dynamics for cooling channel with oblique fin which is the major contribution of the current study. Nevertheless, we have added latest more recent work on cooling channel in the revised manuscript.

·       In the simulation, cases for oblique angle 20, 30, 45, 60 and 90 were tested. It is suggested that the author could give the reason why those cases were selected. Besides, in an actual situation, which oblique angle will be optimal choice?.

The oblique angles are selected to obtain the broad range of possible angle utilized in cooling channel. Based on the obtained result, channels with oblique angle of 90° is the most superior configuration for the cooling purposes: albeit it has high pressure drop, it offers more uniform temperature distribution and lower maximum temperature which is crucial in cooling application. In addition, it generates the lowest entropy which is good from the second thermodynamic law perspective.

·       The current work was performed mainly through numerical methods, and how the author validate the results corresponding actual application.

As presented in the manuscript, we have compared our model with the analytical solution to verify the validity and accuracy of our model. Experimental work for validation purpose is planned in our near future work.

Round  2

Reviewer 3 Report

The problems I have proposed in the last version were carefully revised and the paper seem acceptable.